

# Prehemodialysis arteriovenous access creation is associated with better cardiovascular outcomes in patients receiving hemodialysis: a population-based cohort study

Cheng-Chieh Yen[1], Mei-Yin Liu[2], Po-Wei Chen[3], Peir-Haur Hung[1], Tse-Hsuan Su[4] and Yueh-Han Hsu[1,5,6]

[1] Division of Nephrology, Department of Internal Medicine, Ditmansion Medical Foundation Chia-Yi Christian Hospital, Chia-Yi City, Taiwan
[2] Health Center, Municipal Jingliau Junior High School, Tainan City, Taiwan
[3] Division of Cardiology, Department of Internal Medicine, National Cheng Kung University Hospital, College of Medicine, National Cheng Kung University, Tainan City, Taiwan
[4] Department of Emergency Medicine, Chang Gung Memorial Hospital Linkou, Taoyuan City, Taiwan
[5] Department of Medical Research, China Medical University Hospital, China Medical University, Taichung City, Taiwan
[6] Department of Nursing, Min-Hwei College of Health Care Management, Tainan City, Taiwan

Corresponding authors
Tse-Hsuan Su,
Hsuan.ths@gmail.com
Yueh-Han Hsu,
cych07023@gmail.com

## ABSTRACT

**Background:** Cardiovascular (CV) disease contributes to nearly half of the mortalities in patients with end-stage renal disease. Patients who received prehemodialysis arteriovenous access (pre-HD AVA) creation had divergent CV outcomes.

**Methods:** We conducted a population-based cohort study by recruiting incident patients receiving HD from 2001 to 2012 from the Taiwan National Health Insurance Research Database. Patients' characteristics, comorbidities, and medicines were analyzed. The primary outcome of interest was major adverse cardiovascular events (MACEs), defined as hospitalization due to acute myocardial infarction, stroke, or congestive heart failure (CHF) occurring within the first year of HD. Secondary outcomes included MACE-related mortality and all-cause mortality in the same follow-up period.

**Results:** The patients in the pre-HD AVA group were younger, had a lower burden of underlying diseases, were more likely to use erythropoiesis-stimulating agents but less likely to use renin–angiotensin–aldosterone system blockers. The patients with pre-HD AVA creation had a marginally lower rate of MACEs but a significant 35% lower rate of CHF hospitalization than those without creation (adjusted hazard ratio (HR) 0.65, 95% confidence interval (CI) [0.48–0.88]). In addition, the pre-HD AVA group exhibited an insignificantly lower rate of MACE-related mortality but a significantly 52% lower rate of all-cause mortality than the non-pre-HD AVA group (adjusted HR 0.48, 95% CI [0.39–0.59]). Sensitivity analyses obtained consistent results.

**Conclusions:** Pre-HD AVA creation is associated with a lower rate of CHF hospitalization and overall death in the first year of dialysis.

## INTRODUCTION

End-stage renal disease (ESRD) has become a major public health issue because of its prevalence in more than two million people worldwide and increasing incidence. Compared with general cohorts, patients with ESRD have a higher relative risk of 5-year mortality (*Robinson et al., 2016*). Among the major causes of mortality, cardiovascular (CV) disease contributes nearly half of the events in this population (*Collins et al., 2010*). Therefore, identifying management of CV complications is essential.

Prehemodialysis (pre-HD) care has been proved to ameliorate the outcomes of ESRD patients on HD maintenance (*Baek et al., 2015*; *Bradbury et al., 2007*). Timely creation of arteriovenous access (AVA), such as native fistula or artificial graft, is one of the crucial methods of care planning. It prevents not only the complications from delayed dialysis but also catheter-related infectious events (*Oliver et al., 2004*). However, CV outcomes following pre-HD AVA surgery are currently divergent. Once pre-HD AVA is created, cardiac output increases and leads to functional and structural changes of the heart, lungs, and vasculature (*Guyton & Sagawa, 1961*; *Munclinger et al., 1987*). *London et al. (1993)* reported that the arteriovenous shunt might result in chronic flow overload and cause cardiac hypertrophy. *Nakhoul et al. (2005)* observed that nitric oxygen production was decreased in patients with arteriovenous fistula and contributed to pulmonary hypertension (HTN). *Korsheed, Burton & McIntyre (2009)* and *Korsheed et al. (2011)* reported improved arterial stiffness, better left ventricular ejection fraction, and lesser heart damage after native fistula creation. Variation in laboratory and imaging parameters makes it difficult to predict the clinical outcomes. Several small-scale studies have reported negative clinical CV results following fistula creation (*MacRae et al., 2004*; *Reddy et al., 2017*; *Vizinho et al., 2014*), while a national study from the United States Renal Data System showed that using pre-HD fistula was strongly associated with lower CV mortality (*Wasse, Speckman & McClellan, 2008*). At present, only a few large-scale studies have explored the association between AVA creation and CV-related hospitalization.

Although ESRD is reported to have the highest prevalence in Taiwan compared with other countries, the 5-year survival rate of Taiwanese patients with ESRD seems better (*Robinson et al., 2016*). The Taiwan pre-ESRD pay-for-performance program, involving education and promotion of pre-HD AVA establishment, might have contributed to the higher survival rate (*Lin et al., 2018*). In Taiwan, more than half of the patients undergoing dialysis received access surgery before their first dialysis session, and access creation was completed in more than 80% of them before their chronic dialysis sessions (*Hsu et al., 2018*). Our study investigated the association between timing of AVA creation and CV outcomes in patients who underwent HD. We hypothesized that pre-HD AVA creation improves the CV outcomes of patients undergoing HD.

## MATERIALS AND METHODS

### Data source

We conducted a retrospective cohort study by using the Taiwan National Health Insurance Research Database (NHIRD), which is a national population-based database, provided by Taiwan National Health Insurance (NHI). The NHI is a single-payer, universal and compulsory healthcare program initiated in 1995 and covers 99.9% of Taiwanese residents. In this study, we used a representative subset of one million persons randomly sampled from the 24 million beneficiaries from the Taiwan NHI between 2000 and 2013. No significant difference was observed between the subset and NHIRD in the distribution of sex, age, and average insured payroll-related amount (*National Health Insurance Administration, 2014*). All identities in the NHIRD are encrypted to guarantee patient privacy. This study was approved by the Institutional Review Board of Ditmanson Medical Foundation Chia-Yi Christian Hospital in Taiwan (CYCH-IRB No. 2018054). Informed consent was waived owing to the absence of interference in decision-making processes of medical care.

### Study design, identification, and grouping of study subjects

We identified patients with chronic kidney disease (CKD) who began HD sessions during 2001–2012 by using the NHI procedure codes of receiving HD. The day of first HD session was employed as the index date. CKD was defined as patients receiving at least two outpatient diagnoses according to International Classification of Diseases, 9th Revision, Clinical Modification (ICD-9-CM) codes within the 1 year prior to the index date. Patients were excluded if they were aged <20 years, had ever received peritoneal dialysis, or had ever received kidney transplantation before or during their first year of HD. We combined patients receiving native fistula and artificial graft for analysis because of their similar CV results (*Ravani et al., 2013*). Pre-HD AVA was defined as its creation date ≥1 month before the index date. Patients in whom AVA was created <1 month prior to HD were excluded owing to their inappropriate access usage according to the guidelines (*Vascular Access Work Group, 2006*; *Ishani et al., 2014*). We further excluded patients who received implantation of HD catheters, namely tunneled and nontunneled catheters, before the index date.

### Data and definitions of study variables

We analyzed the characteristics, comorbidities, and medicines of the included patients. Owing to the NHI charged its beneficiaries different amounts of insurance premiums according to their earnings, the socioeconomic status of patients was represented by their income, which was obtained according to the average insured payroll-related amounts. Comorbidities were defined as patients experiencing at least one hospitalization or two outpatient visits, which expressed in terms of the corresponding ICD-9-CM codes of any of the following illnesses, within the 1 year prior to the index date: HTN, ischemic heart disease (IHD), congestive heart failure (CHF), cerebrovascular accident (CVA), peripheral vascular disease (PVD), dysrhythmia, diabetes mellitus (DM), chronic obstructive pulmonary disease (COPD), peptic ulcer disease, liver disease, cancer, and

dementia. Additionally, to denote the disease burdens we applied the Taiwan index, which is a comorbidity index for mortality prediction validated for Taiwanese patients with incident HD (*Chen et al., 2014*). Medicines including erythropoiesis-stimulating agent (ESA), antiplatelet agent, anticoagulant agent, angiotensin converting enzyme inhibitor (ACEI), angiotensin II receptor blocker, and statin were defined as more than two prescriptions during ambulatory visits within the 1 year prior to the index date and was expressed in terms of the anatomical therapeutic chemical classification system.

## Outcomes of study subjects

Primary outcome of the present study was major adverse cardiovascular events (MACEs), which was defined as ICD-9-CM-based hospitalization for acute myocardial infarction, CVA, or CHF that occurred within the 1 year after the first HD session. Secondary outcomes were all-cause mortality, MACE-related and bloodstream infection (BSI)-related mortality, which was defined as overall death and death resulting from MACE and BSI in corresponding ICD-9-CM codes, within the same follow-up period following the first dialysis. To validate the findings, we performed multiple sensitivity analyses including exclusion of patients without AVA creation, exclusion of patients not receiving regular HD, and inclusion of patients receiving AVA creation within 1 month.

## Statistical analysis

We compared patient characteristics, comorbidities, and medical prescriptions between the pre-HD AVA and non-pre-HD AVA groups. Continuous data were reported as a median or mean and were analyzed using the Mann–Whitney test or independent *t*-test as appropriate. Categorical data were reported as percentages and were analyzed using the Chi-squared test. Since management of CKD changed over time, we included the year of starting HD as a time indicator. We constructed our propensity score model using variables related to patient characteristics, year of HD, comorbidities, and medical prescriptions, and calculated model using the logistic regression method. We performed 1:1 matching using the nearest neighbor algorithm without replacement and with a 0.05 caliper width to reduce imbalances between groups. After matching, we used absolute standardized differences (ASDs) to evaluate the balance between groups. An ASD threshold of 10% was used to delineate good and poor balance (*Austin, 2008*).

We evaluated the cause-specific mortality and overall mortality over 1 year using weighted cumulative incidence curves considering competing risks of death. Event-time was measured from the index date until the date of the event, 1 year after the index date, or the end of the study (December 31, 2013), whichever occurred earlier. We applied the Fine–Gray subdistributional hazard models to calculate crude and adjusted hazard ratios (HRs). Robust sandwich variance was used to correct the correlated data structure after matching (*Austin, 2014*). Because the 1:1 propensity score matching would reduce the sample size, we performed the inverse probability treatment weighting (IPTW) of study subjects as another sensitivity analysis. Since extreme weights might cause bias estimate in the marginal hazard ratio, we used IPTW in the subsample restricted to patients with a propensity score between 0.1 and 0.9 (*Austin & Stuart, 2017*). A 2-tailed

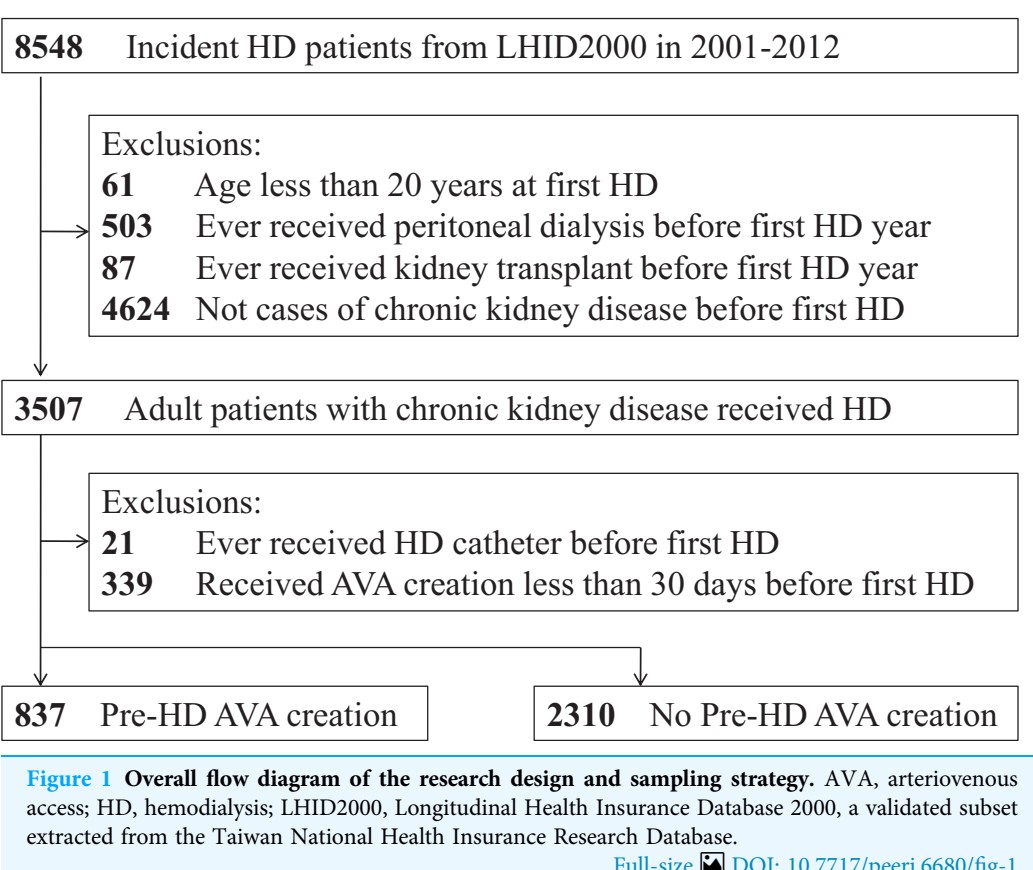

**Figure 1 Overall flow diagram of the research design and sampling strategy.** AVA, arteriovenous access; HD, hemodialysis; LHID2000, Longitudinal Health Insurance Database 2000, a validated subset extracted from the Taiwan National Health Insurance Research Database.

$p$-value of < 0.05 indicated statistical significance. SAS version 9.4 (SAS Institute, Inc., Cary, NC, USA) was used for analyses.

## RESULTS

Figure 1 illustrates the research design and sampling strategy. Overall, this study analyzed 3,147 patients—837 patients (26.6%) receiving pre-HD AVA creation and 2,310 patients (73.4%) not receiving pre-HD AVA creation.

The baseline characteristics of the recruited patients are presented in Table 1. The median age of patients with pre-HD AVA creation was lower than that of those without (66 years vs. 71 years, ASD: 29.9%). No statistically significant difference was observed in terms of sex between the two groups. Considering comorbidities, the patients in the pre-HD AVA group had lower proportions of CVA, dysrhythmia, COPD, liver disease, and dementia than those in the non-pre-HD AVA group. The groups exhibited similar proportions of patients with IHD, CHF, PVD, and DM. The mean Taiwan index of pre-HD AVA group was significantly lower than that of the non-pre-HD AVA group (5.55 ± 4.0 vs. 6.55 ± 4.4, ASD: 23%). Additionally, more patients with pre-HD AVA creation received ESAs than did those without pre-HD AVA creation (81.72% vs. 34.81%, ASD: 108%), whereas less patients with pre-HD AVA received ACEIs and ARBs than did those without pre-HD AVA (60.57% vs. 68.35%, ASD: 16.3%). The prescription of antiplatelet agent, anticoagulant agent, or statin was similar among the groups.

**Table 1 Characteristics of patients with and without prehemodialysis arteriovenous access.**

| | Unmatched | | | Matched | | |
|---|---|---|---|---|---|---|
| | Pre-HD AVA (N = 837) | No pre-HD AVA (N = 2,310) | ASD (%) | Pre-HD AVA (N = 792) | No pre-HD AVA (N = 792) | ASD (%) |
| Age, years | | | | | | |
| Median (IQR) | 66 (56–74) | 71 (59–79) | 29.9* | 67 (56–75) | 66 (55–75) | 2.42 |
| Sex, male (%) | 440 (52.57) | 1,157 (50.09) | 4.96 | 401 (50.63) | 423 (53.41) | 5.56 |
| Income, NTD/year | | | 15.3* | | | 1.83 |
| Dependent | 296 (35.36) | 878 (38.01) | | 284 (35.86) | 288 (36.36) | |
| 1–19,999 | 206 (24.61) | 700 (30.30) | | 197 (24.87) | 197 (24.87) | |
| 20,000–39,999 | 275 (32.86) | 645 (27.92) | | 259 (32.70) | 261 (32.95) | |
| ≥40,000 | 60 (7.17) | 87 (3.77) | | 52 (6.57) | 46 (5.81) | |
| Comorbidities | | | | | | |
| HTN | 712 (85.07) | 1,899 (82.21) | 7.73 | 670 (84.60) | 673 (84.97) | 1.05 |
| IHD | 218 (26.05) | 649 (28.10) | 4.61 | 197 (24.87) | 200 (25.25) | 0.87 |
| CHF | 393 (46.95) | 1,150 (49.78) | 5.66 | 364 (45.96) | 364 (45.96) | 0.00 |
| CVA | 76 (9.08) | 363 (15.71) | 20.2* | 75 (9.47) | 76 (9.60) | 0.42 |
| PVD | 45 (5.38) | 152 (6.58) | 5.07 | 40 (5.05) | 47 (5.93) | 3.88 |
| Dysrhythmia | 50 (5.97) | 226 (9.78) | 14.1* | 49 (6.19) | 44 (5.56) | 2.68 |
| DM | 461 (55.08) | 1,303 (56.41) | 2.60 | 429 (54.17) | 440 (55.56) | 2.79 |
| COPD | 81 (9.68) | 379 (16.41) | 20.0* | 78 (9.85) | 82 (10.35) | 1.67 |
| PUD | 186 (22.22) | 569 (24.63) | 5.69 | 175 (22.10) | 172 (21.72) | 0.91 |
| Liver disease | 65 (7.77) | 252 (10.91) | 10.0* | 65 (8.21) | 56 (7.07) | 4.27 |
| Cancer | 74 (8.84) | 233 (10.09) | 4.25 | 71 (8.96) | 69 (8.71) | 0.80 |
| Dementia | 20 (2.39) | 124 (5.37) | 15.4* | 20 (2.53) | 17 (2.15) | 2.50 |
| Taiwan index (mean ± SD) | 5.55 ± 4.00 | 6.55 ± 4.40 | 23.0* | 5.52 ± 4.02 | 5.52 ± 4.04 | 0.09 |
| Medicine | | | | | | |
| ESAs | 684 (81.72) | 804 (34.81) | 108.* | 639 (80.68) | 635 (80.18) | 1.27 |
| Antiplatelets | 537 (64.16) | 1,563 (67.66) | 7.30 | 505 (63.76) | 505 (63.76) | 0.00 |
| Anticoagulants | 75 (8.96) | 161 (6.97) | 7.35 | 53 (6.69) | 51 (6.44) | 1.00 |
| ACEI/ARBs | 507 (60.57) | 1,579 (68.35) | 16.3* | 485 (61.24) | 500 (63.13) | 3.90 |
| Statins | 262 (31.30) | 665 (28.79) | 5.48 | 244 (30.81) | 247 (31.19) | 0.81 |

Notes:
Income was divided into four strata according to insurance fees: dependent (patient's medical expenditure was taken charge of the government), <20,000 New Taiwan Dollars (NTD) per month, 20,000–40,000 NTD per month, and >40,000 NTD per month.
The Taiwan index is a weighted comorbidity score of IHD × 1 + CHF × 3 + CVA × 4 + PVD × 2 + COPD × 3 + PUD × 2 + Liver disease × 4 + Dysrhythmia × 3 + Cancer × 6 + DM × 3.
ACEI, angiotensin converting enzyme inhibitor; ARB, angiotensin II receptor blocker; ASD, absolute standardized difference; CHF, congestive heart failure; COPD, chronic obstructive pulmonary disease; CVA, cerebrovascular accident; DM, diabetes mellitus; ESA, erythropoiesis-stimulating agent; HTN, hypertension; IHD, ischemic heart disease; IQR, interquartile range; Pre-HD AVA, prehemodialysis arteriovenous access; PUD, peptic ulcer disease; PVD, peripheral vascular disease; SD, standard deviation.
* ASD ≥ 10%.

Table 2 shows the primary and secondary outcomes of the pre-HD AVA and non-pre-HD AVA groups. The patients with pre-HD AVA creation had a lower rate of MACEs during the follow-up period than did those without pre-HD AVA creation (crude HR 0.73, 95% CI [0.6–0.89]), but the effect became nonsignificant after matching for

**Table 2 Clinical outcomes of patients with and without prehemodialysis arteriovenous access.**

| | Unmatched | | Matched | |
|---|---|---|---|---|
| | Crude HR | Adjusted HR[a] | Crude HR | Adjusted HR[b] |
| **Primary outcomes** | | | | |
| MACEs | 0.73 (0.60–0.89)** | 0.89 (0.71–1.11) | 0.93 (0.73–1.18) | 0.94 (0.74–1.21) |
| CHF | 0.52 (0.40–0.68)*** | 0.65 (0.48–0.88)** | 0.63 (0.46–0.86)** | 0.63 (0.46–0.87)** |
| **Secondary outcomes** | | | | |
| All-cause mortality | 0.28 (0.23–0.34)*** | 0.48 (0.39–0.59)*** | 0.47 (0.37–0.60)*** | 0.46 (0.36–0.59)*** |
| MACE-related mortality | 0.37 (0.25–0.55)*** | 0.70 (0.45–1.08) | 0.59 (0.37–0.93)* | 0.60 (0.38–0.96)* |
| BSI-related mortality | 0.21 (0.15–0.30)*** | 0.32 (0.22–0.46)*** | 0.30 (0.20–0.45)*** | 0.29 (0.19–0.44)** |

**Notes:**
[a] Adjusted for age, sex, income, year of hemodialysis, comorbidities, and medicines;
[b] adjusted for age, sex, income, year of hemodialysis, Taiwan index, hypertension, dementia, and medicines. The Taiwan index is a comorbidity index employed for mortality prediction that has been validated for Taiwanese patients undergoing hemodialysis as having adequate reclassification ability.
BSI, bloodstream infection; CHF, congestive heart failure; HR, hazard ratio; MACEs, major adverse cardiovascular events.
* $p < 0.05$;
** $p < 0.01$;
*** $p < 0.001$.

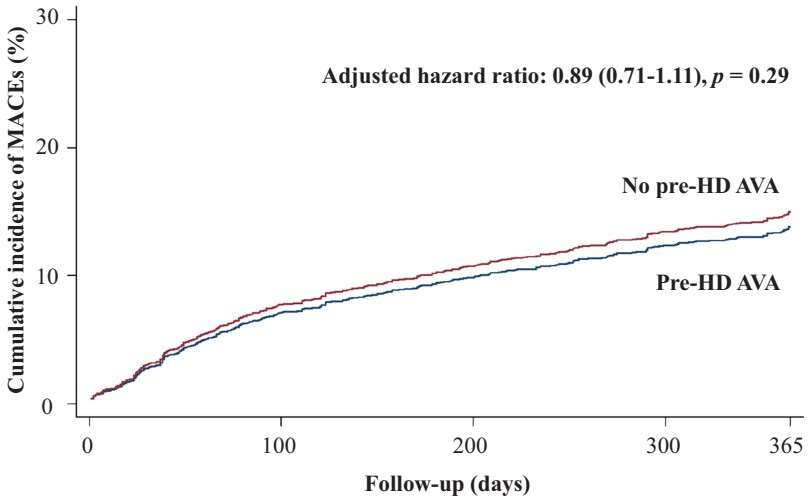

**Figure 2 Cumulative incidence of major adverse cardiovascular events in patients with and without prehemodialysis arteriovenous access creation.** MACEs, major adverse cardiovascular events; Pre-HD AVA, prehemodialysis arteriovenous access.

age, sex, year of HD, comorbidities, and medicines (adjusted HR 0.89, 95% CI [0.71–1.11], $p = 0.29$, Fig. 2). We further analyzed the MACEs separately and observed that patients with pre-HD AVA creation had a 35% lower CHF hospitalization rate after matching (adjusted HR 0.65, 95% CI [0.48–0.88], $p < 0.01$, Fig. 3). We examined CHF hospitalization rate in the propensity-score-matched groups, which showed a similar trend (adjusted HR 0.63, 95% CI [0.46–0.87], $p < 0.01$).

Regarding secondary outcomes, we revealed that patients in the pre-HD AVA group had a 52% lower rate of all-cause mortality (adjusted HR 0.48, 95% CI [0.39–0.59], $p < 0.001$, Fig. 4), a marginally lower rate of MACE-related mortality (adjusted HR 0.7, 95% CI [0.45–1.08], $p = 0.1$, Fig. 5), and a 68% lower rate of BSI-related mortality

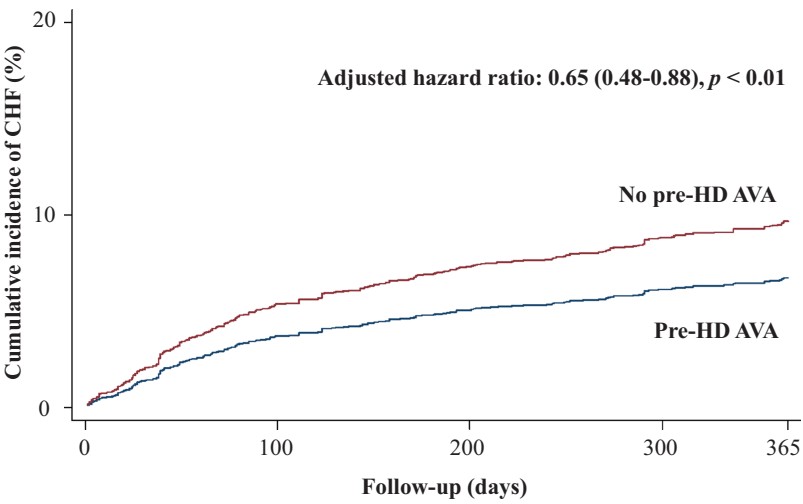

**Figure 3 Cumulative incidence of congestive heart failure in patients with and without prehemodialysis arteriovenous access creation.** CHF, congestive heart failure; Pre-HD AVA, prehemodialysis arteriovenous access.

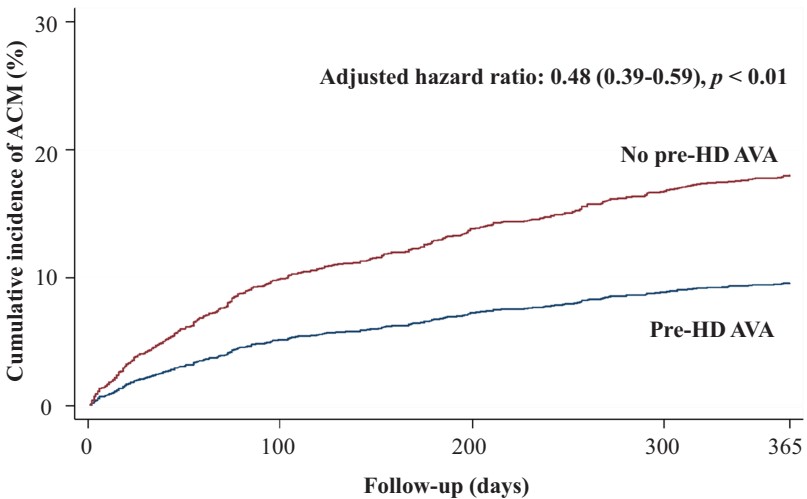

**Figure 4 Cumulative incidence of all-cause mortality in patients with and without prehemodialysis arteriovenous access creation.** ACM, all-cause mortality; Pre-HD AVA, prehemodialysis arteriovenous access.

(adjusted HR 0.32, 95% CI [0.22–0.46], $p < 0.001$, Fig. S1) than those in the non-pre-HD AVA group. We next examined the outcomes in the propensity-score-matched groups. The patients with pre-HD AVA creation exhibited a 54% lower rate of all-cause mortality (adjusted HR 0.46, 95% CI [0.36–0.59], $p < 0.001$), a 40% lower rate of MACE-related mortality (adjusted HR 0.60, 95% CI [0.38–0.96] , $p < 0.05$), and a 71% lower rate of BSI-related mortality (adjusted HR 0.29, 95% CI [0.19–0.44], $p < 0.01$) than those without pre-HD AVA creation.

Sensitivity analyses of the outcomes are presented in Table 3. They revealed a consistently lower CHF hospitalization rates, ranging from 29% to 43%, of patients

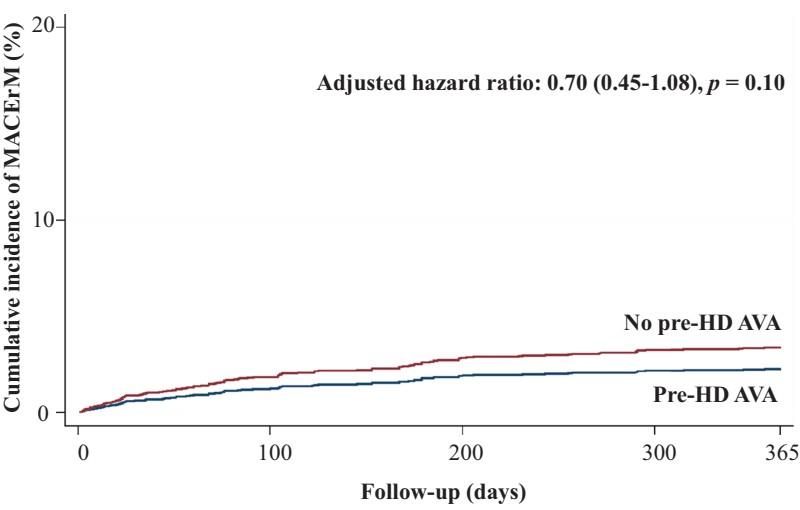

**Figure 5 Cumulative incidence of major adverse cardiovascular event-related mortality in patients with and without prehemodialysis arteriovenous access creation.** MACErM, major adverse cardiovascular event-related mortality; Pre-HD AVA, prehemodialysis arteriovenous access.

**Table 3 Sensitivity analyses of clinical outcomes of patients with and without prehemodialysis arteriovenous access.**

|  | SA1[b] | SA2[c] | SA3[d] | SA4[e] |
|---|---|---|---|---|
| Primary outcomes |  |  |  |  |
| MACEs, aHR[a] | 0.77 (0.62–0.97)* | 0.82 (0.64–1.05) | 0.90 (0.74–1.10) | 0.88 (0.69–1.13) |
| CHF, aHR | 0.57 (0.42–0.77)*** | 0.60 (0.43–0.84)** | 0.69 (0.53–0.91)** | 0.71 (0.51–0.98)* |
| Secondary outcomes |  |  |  |  |
| All-cause mortality, aHR | 0.94 (0.72–1.22) | 0.65 (0.46–0.91)* | 0.50 (0.42–0.60)*** | 0.47 (0.36–0.61)*** |
| MACE-related mortality, aHR | 0.93 (0.56–1.56) | 0.79 (0.42–1.49) | 0.65 (0.43–0.97)* | 0.70 (0.43–1.17) |
| BSI-related mortality, aHR | 0.60 (0.39–0.92)* | 0.38 (0.21–0.67)*** | 0.34 (0.25–0.47)*** | 0.29 (0.19–0.46)*** |

Notes:
[a] Adjusted for age, sex, income, year of hemodialysis, comorbidities, and medicines;
[b] exclusion of patients not receiving arteriovenous access in the first year of dialysis;
[c] exclusion of patients not receiving regular hemodialysis;
[d] inclusion of patients receiving arteriovenous access less than 1 month before the first hemodialysis;
[e] inverse probability of treatment weighting of study subjects.
aHR, adjusted hazard ratio; BSI, bloodstream infection; CHF, congestive heart failure; MACEs, major adverse cardiovascular events; SA, sensitivity analysis.
* $p < 0.05$;
** $p < 0.01$;
*** $p < 0.001$.

receiving pre-HD AVA creation. In addition, they exhibited a consistently lower BSI-related mortality rates of patients in the pre-HD AVA group.

## DISCUSSION

In our nationally representative cohort, we observed that patients with pre-HD AVA creation had a 35% lower CHF hospitalization rate and 52% lower all-cause mortality rate than those without during the first year of HD—significant differences. Additionally, we disclosed a marginally lower rate of MACEs and MACE-related mortality during the same follow-up period. Pre-HD AVA creation might be associated with better CV outcomes within the first year of HD.

In this study, the patients with pre-HD AVA creation were younger, had a lesser burden of comorbidities, had a higher percentage of ESAs but a lower percentage of ACEIs and ARBs usage (Table 1). Age is a well-established factor affecting postsurgery prognosis. The significantly lower Taiwan index of the pre-HD AVA group indicated the lower disease burden of the patients receiving pre-HD AVA compared with those not receiving pre-HD AVA. We also observed that the patients without pre-HD AVA creation were significantly more likely to have CVA, dysrhythmia, COPD, or liver disease than those with pre-HD AVA creation, and those diseases might reflect higher neurologic, respiratory, and coagulatory risks during the operation. In one previous study, patients with dementia had a greater risk of early death and fatal complications postoperatively (*Kassahun, 2018*). Thus, patients who were young or had fewer comorbidities were willing to undergo pre-HD AVA surgery. Additionally, a different proportion of patients receiving ESAs and renin–angiotensin–aldosterone system blockers might imply more recruitment of pre-HD care, which promotes the possibility of dialysis access creation by education (*Ishani et al., 2014*).

Our study revealed a significantly lower CHF hospitalization rate within the first year of HD among the patients receiving pre-HD AVA creation (Fig. 3). Consistent results were obtained for the other matched models (Table 2) and sensitivity analyses (Table 3). Patients who undergo AVA surgery before HD might avoid delayed HD, thus preventing exacerbated fluid overload and increased CHF risk. The increased cardiac preload after AVA surgery is compensated by a corresponding decrease in peripheral vascular resistance following surgery (*Ori et al., 1996*) and consequent fluid removal during HD sessions. The fluid status of most patients undergoing HD has been proved to achieve a new balance shortly (*Alkhouli et al., 2015*; *Dal Canton et al., 1981*). In the Dialysis Outcomes and Practice Patterns Study, *Rayner et al. (2003)* observed a low flow rate of the fistula in Japanese patients, which might be related to low-caliber vessels in the Asian population. The degree of CV damage due to blood volume following AVA creation might differ according to vascular characteristics. Several studies have supported our findings of CV benefits following pre-HD AVA creation: *Ori et al. (1996)* conducted an echocardiographic study to observe cardiac performance before and after AVA creation. A gentle volume overload developed postoperatively but was offset by decreased vascular resistance. The shortening and ejection fractions of the left ventricle improved 2 weeks after the AVA operation. *Sandhu et al. (2004)* observed that none of 17 patients receiving native fistula before HD developed CHF during the 6-week-period following surgery. They concluded that the postoperative changes in cardiac index, stroke volume, and vascular resistance were physically minimal and without extra loading of patients' hemodynamics. Thus, CHF might not occur or worsen after AVA creation. Further investigation is warranted to clarify the causality of AVA in CV outcomes.

Some studies have obtained contrasting results from ours: *MacRae et al. (2004)* noted that a patient undergoing HD developed cardiac failure under a high-flow arteriovenous fistula and concluded that the high fistula flow caused myocardium decompensation with a decline in the ejection fraction. Other studies have adopted an opposite viewpoint on ejection fraction alteration following AVA creation (*Iwashima et al., 2002*;

*Korsheed et al., 2011*; *Ori et al., 1996*). *Vizinho et al. (2014)* reported that pre-HD AVA creation was associated with a decrease in the subendocardial viability ratio, which predicted a poorer outcome regarding CV hospitalization. Nevertheless, a small sample size and lack of CV comorbidity adjustments limit the relevance of their speculation. *Reddy et al. (2017)* traced CV changes of patients following native shunt creation for 2.6 years and observed that remodeling and dysfunction of the right ventricle developing after shunt operation and dialysis initiation caused increased risks of CHF and death. However, the absence of controls and uncertainty in the effect of AVA and dialysis on cardiac dysfunction make the supposition inconclusive. More large-scale and close-matching studies should be planned to confirm the relationship.

Considering secondary outcomes, we evaluated the effect of pre-HD AVA creation on overall mortality and disclosed a 52% lower rate of all-cause mortality in the pre-HD AVA group (Table 2; Fig. 4). In addition to CV disease, catheter-associated infectious disease, mainly those transmitted through the bloodstream, is another major cause of mortality in patients undergoing dialysis. We assumed that the lower all-cause mortality might have been related to the 68% reduction in the rate of BSI-related mortality in the pre-HD AVA group (Table 2; Fig. S1), which was due to lesser usage of HD catheters. Furthermore, we also observed that the patients with pre-HD AVA creation had an insignificantly lower rate of MACE-related mortality compared with those without after propensity score matching (Table 2; Fig. 5). We believed that this type of mortality was mainly affected by the underlying diseases of patients rather than AVA surgery. The literature has suggested that HTN, IHD, CHF, and DM influence the CV mortality rate for patients undergoing HD (*Banerjee et al., 2007*; *Lee et al., 2016*; *Zoccali, Tripepi & Mallamaci, 2005*). Because our groups had similar distributions of these diseases, our finding was in fair agreement with the literature.

This was a country-based study including all pre-HD patients underlined CKD matched for age, sex, income, year of HD, comorbidities, and associated medicines. In Taiwan, the NHI Bureau has launched a pay-for-performance program focusing on patients of glomerulus filtration rate <45 ml/min/1.73 $m^2$ from 2006. The incentive payment from the bureau is paid to medical institutions if their recruited patients achieved the targets on blood pressure, glycated hemoglobin, nursing education, nutrition consult, and so on. Patients with ESRD have received comprehensive access evaluation by qualified nephrologists at outpatient clinics or during admissions. Most studies exploring the effects of pre-HD AVA creation on CV outcomes compared associated parameters before and after surgery (*Dal Canton et al., 1981*; *Dundon et al., 2014*; *Iwashima et al., 2002*; *Korsheed et al., 2011*; *Munclinger et al., 1987*; *Ori et al., 2002, 1996*; *Reddy et al., 2017*; *Sandhu et al., 2004*; *Savage et al., 2002*; *Utescu et al., 2009*; *Vizinho et al., 2014*). In addition, some compared the effects before dialysis initiation to exclude the impact of dialysis on CV performance (*Dundon et al., 2014*; *Iwashima et al., 2002*; *Korsheed et al., 2011*; *Ori et al., 1996*; *Savage et al., 2002*). However, selection bias would have been unavoidable in these studies because patients with pre-HD AVA creation tend to be compliant in medical practice, which would affect their overall outcomes. Furthermore, excluding the dialysis effect appears impractical considering the goal of AVA preparation. Moreover, dialysis is a

well-known risk factor of cardiac injury, and its vintage is positively associated with the degree of injury (*McIntyre, 2009*). Once the AVA was used for dialysis, evaluating the CV prognoses in combination with HD was difficult. We followed up for 1 year after HD initiation because we assumed that the effect of pre-HD AVA creation would be offset by a longer period of HD. Our study provides another perspective regarding evaluation of the benefits and hazards of pre-HD AVA surgery.

This study had several limitations. First, the present study was an observational study, which non-observed confounders might restrict the inference. Second, the NHIRD is an administrative database in which the identification of comorbidities is based solely on ICD-9-CM codes rather than clinical criteria; misclassifications might thus have occurred, leading to residual confounding. Additionally, we recruited our patients until the end of 2012 since the data was valid until December 31, 2013 in LHID 2000. Third, patients could choose the preferred medical providers for AVA creation or HD freely owing to high medical accessibility in Taiwan. It is difficult to determine the relationship between patients and medical providers, which influenced the timing of AVA creation. Fourth, the indications of CHF hospitalization are varied among patients and medical facilities. However, the NHIRD does not provide objective parameters of cardiac alteration, such as the level of natriuretic peptide, ejection fraction of ventricles, or pulse wave velocity of vessels, which could support our findings. Fifth, our study did not consider medicines such as calcium channel blockers, beta blockers, or diuretics, which have been shown to influence CV outcomes in patients undergoing HD (*Georgianos & Agarwal, 2016*; *Karaboyas et al., 2018*). Lastly, it is difficult to distinguish the absolute effect of pre-HD AVA on CV outcomes in combination with personal compliance and dialysis factors affecting the CV system. Integrated trials comprising data and imaging should be further conducted to corroborate our results.

## CONCLUSIONS

In this population-based cohort study, patients with pre-HD AVA creation had a 35% lower CHF hospitalization rate and a 52% lower all-cause mortality rate than those without pre-HD AVA creation within the first year of HD. Marginal benefits were also observed in terms of MACEs and MACE-related mortality during the same follow-up period. Pre-HD AVA creation might be associated with better CV outcomes in the first year of HD and should be promoted in the pre-HD care focusing on patients with late-stage CKD.

## ACKNOWLEDGEMENTS

We would like to thank Ching-Fang Tsai for the pretest of study design, and Professor Chih-Cheng Hsu for his experienced dehortation and encouragement throughout the study.

### Funding

The authors received no funding for this work.

## Competing Interests

The authors declare that they have no competing interests.

## Author Contributions

- Cheng-Chieh Yen conceived and designed the experiments, analyzed the data, contributed reagents/materials/analysis tools, prepared figures and/or tables, authored or reviewed drafts of the paper, approved the final draft.
- Mei-Yin Liu conceived and designed the experiments, analyzed the data, authored or reviewed drafts of the paper.
- Po-Wei Chen authored or reviewed drafts of the paper.
- Peir-Haur Hung authored or reviewed drafts of the paper.
- Tse-Hsuan Su analyzed the data, contributed reagents/materials/analysis tools, prepared figures and/or tables, authored or reviewed drafts of the paper, approved the final draft.
- Yueh-Han Hsu authored or reviewed drafts of the paper, approved the final draft.

## Human Ethics

The following information was supplied relating to ethical approvals (i.e., approving body and any reference numbers):

This study was approved by the Institutional Review Board of Ditmanson Medical Foundation Chia-Yi Christian Hospital in Taiwan (CYCH-IRB No. 2018054).

## Data Availability

We identified patients with chronic kidney disease who began hemodialysis sessions from 2001 to 2012 in the validated subgroup extracted from the Taiwan National Health Insurance Research Database, this data was provided for peer review.

The raw data for this study was obtained by application to the Taiwan National Health Insurance Research Database (http://nhird.nhri.org.tw/en/index.htm) and may not be shared according to their regulations. Access to data in the Database may be granted to eligible Taiwanese citizens who fulfill the requirements for conducting research projects (http://nhird.nhri.org.tw/en/Data_Protection.html).

## Supplemental Information

Supplemental information for this article can be found online at http://dx.doi.org/10.7717/peerj.6680#supplemental-information.

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
