# Peer review of "Prehemodialysis arteriovenous access creation is associated with better cardiovascular outcomes in patients receiving hemodialysis: a population-based cohort study"

_PeerJ, doi:10.7717/peerj.6680_

## Round 0.1 · original submission · Major Revisions

Your paper was reviewed by 2 independent experts in this field and there are some revisions which will be needed. In particular Reviewer 2 has raised several issues in the Experimental Design and Reporting which should be thoroughly addressed in any subsequent revisions. Because of the nature of this study, some methodological issues the Reviewers pointed out may not be able to be remedied, however at a minimum these should be clearly acknowledged and delineated under the "Limitations" section of the Discussion.

·

Basic reporting

OK

Experimental design

OK

Validity of the findings

OK

Additional comments

The study is well designed and the analysis looks appropriate.

Reviewer 2 ·

Basic reporting

This is a potentially interesting observational study regarding the timing of AVA implant in dialysis patients, however, the study suffers from several limitations. The research question is very simplified. There seems to be evidence that the implant prior to 1 month from dialysis date has evidence for efficacy as reflected in guidelines which were not discussed in the introduction. There is strong selection bias which may not be attenuated by propensity matching, how about provider quality which may impact implant timing and mortality. Also, the study did not control relevant factors such as ACE/ARBs use, statins, time effect which could impact CVD mortality. I have included additional comments in relevant sections.

Several of the sentences are hard to read and some have grammatical errors.

Line 91 – are dates in the dataset, date-shifted? Date of service, day of first HD session or hospitalization are considered within the 18 HIPAA identifiers.

In the supplementary data, are admission/service and event dates, date-shifted? Service dates are considered personally identifiable information and should not be shared in a de-identified dataset or may be shared by date shifting with an unknown interval at least >45 days. Need to discuss with their local IRB.

Experimental design

Line 79-81 – there is threat of selection bias in this study. There could be other reasons that some patients received this service earlier on (higher income, went to a better provider, etc.) and which could impact their CVD event and mortality outcomes. The propensity method won’t attenuate selection bias due to non-observed confounders – it will only make the two groups comparable on the observed confounders.

Lines 104-105 – AVA implanted <1month prior to the index date of HD were excluded, this brings a very important point regarding the study design. It seems there is enough evidence as reflected by guidelines that AVA implant before 1 month from HD is efficacious. This was not included in the introduction. Also, it brings to the study question regarding when is the best timing to implant AVA. A better design would be to treat time as continuous with an indicator of pre and post dialysis, because the study design as in the current format, excluding this arm from the study leads to missing information. Also, what is the implications for practice – is it lack of knowledge from providers as to when they predict the patient to start dialysis or when to implant the AVA.

Line 110 – which definition of comorbidities did you use, the HCC or US CMS’s data warehouse, other?

Line 118 – why did you pick those medicines. How about statins, ACE/ARBs, or diabetes meds? Statins, ACE/ARBS and diabetes meds may impact CVD events and mortality.

Events were observed only one year after index date – therefore, why conduct the study using data from earlier years, why not to extend the study to more recent years.

Consider using an alternative methods of propensity score matching, like inverse proportion, as to not exclude participants from the study, why doing 1 to 1 match? Did you include income in the propensity score matching?

Did you include time effects in the models since the management of CKD and dialysis has changes over time.

The statistical analysis section is very poorly described, consider adding better description about the actual statistical analysis tests and details about the models, which statistical tests were used and in which program.

Validity of the findings

The study suffers from several limitations described previously which would dampen enthusiasm about the findings.

Line 155, the use of the word obvious is ambiguous, use instead either statistically or clinically significant or not.

Additional comments

These are additional specific comments that I did not include in the previous sections.

When was the P4P program established, and what is the structure, is it incentive payment for nephrologists, primary care docs or medical teams/institutions?

Introduction

Line 54, this sentence is not clear, do you mean higher relative risk of 5-year mortality or higher percent 5-year mortality compared to which % in the general population?

Need to define what is AVA, and that AVA is arteriovenous shunt or fistula before using the terms interchangeably.

Line 71 – which other large-scale studies other than the USRDS study that was included?

Line 90 – what is the average insured payroll-related amount, is it what they pay for the insurance and indicator of the participant socioeconomic status?

The discussion section is a repetition of the results. It should be modified as to describe how the results may impact clinical practice and related to the guidelines they mentioned in the exclusion section. Also, what are the recommendations regarding the P4P program.

The issue around provider timing of AVA implant and why some providers do or do not implant by certain time is not well discussed, if it's only patient characteristics that impact timing or are there other provider or health system level factors?

---

## Round 0.2 · Minor Revisions

Please incorporate your responses (contained in your rebuttal letter) to the following Reviewer queries into the appropriate section (Introduction, Methods, Discussion, Limitations, etc). :

04. Lines 104-105 – AVA implanted <1month prior .... (Note: Methods section is acceptable)
06. Also, what is the implications for practice... (Clinical implications can be noted in Discussion)
09. Events were observed only one year after index date....(Limitations section)
14. When was the P4P program established...(Methods or Discussion section is acceptable)
17. Line 71 – which other large-scale studies...(Methods or Limitations section is acceptable)
18. Line 90 – what is the average insured payroll-related amount... (Methods section is acceptable)
20. The issue around provider timing of AVA implant....Limitations section)

---

## Round 0.3 · accepted · Accept

Your revisions have adequately addressed the substantive issues raised.